# Cadmium Disrupted ER Ca^2+^ Homeostasis by Inhibiting SERCA2 Expression and Activity to Induce Apoptosis in Renal Proximal Tubular Cells

**DOI:** 10.3390/ijms24065979

**Published:** 2023-03-22

**Authors:** Kongdong Li, Chuanzhi Guo, Jiacheng Ruan, Bo Ning, Chris Kong-Chu Wong, Haifeng Shi, Jie Gu

**Affiliations:** 1School of Life Sciences, Jiangsu University, Zhenjiang 212013, China; 2Department of Biology, Hong Kong Baptist University, Hong Kong SAR, China

**Keywords:** ER stress, Ca^2+^, apoptosis, cadmium

## Abstract

Cadmium (Cd^2+^) exposure induces chronic kidney disease and renal cancers, which originate from injury and cancerization of renal tubular cells. Previous studies have shown that Cd^2+^ induced cytotoxicity by disrupting the intracellular Ca^2+^ homeostasis that is physically regulated by the endoplasmic reticulum (ER) Ca^2+^ store. However, the molecular mechanism of ER Ca^2+^ homeostasis in Cd^2+^-induced nephrotoxicity remains unclear. In this study, our results firstly revealed that the activation of calcium-sensing receptor (CaSR) by NPS R-467 could protect against Cd^2+^ exposure-induced cytotoxicity of mouse renal tubular cells (mRTEC) by restoring ER Ca^2+^ homeostasis through the ER Ca^2+^ reuptake channel sarco/endoplasmic reticulum Ca^2+^-ATPase (SERCA). Cd^2+^-induced ER stress and cell apoptosis were effectively abrogated by SERCA agonist CDN1163 and SERCA2 overexpression. In addition, in vivo, and in vitro results proved that Cd^2+^ reduced the expressions of SERCA2 and its activity regulator phosphorylation phospholamban (p-PLB) in renal tubular cells. Cd^2+^-induced SERCA2 degradation was suppressed by the treatment of proteasome inhibitor MG132, which suggested that Cd^2+^ reduced SERCA2 protein stability by promoting the proteasomal protein degradation pathway. These results suggested that SERCA2 played pivotal roles in Cd^2+^-induced ER Ca^2+^ imbalance and stress to contribute to apoptosis of renal tubular cells, and the proteasomal pathway was involved in regulating SERCA2 stability. Our results proposed a new therapeutic approach targeting SERCA2 and associated proteasome that might protect against Cd^2+^-induced cytotoxicity and renal injury.

## 1. Introduction

Cadmium (Cd), as a widely distributed environmental pollutant, is unavoidably absorbed by humans through environmental and occupational exposures [1]. After intake into the human body, approximately 50% Cd as the divalent cation (Cd^2+^) accumulates in the kidney, in which the renal proximal tubule is the first opportunistic site of Cd^2+^ reabsorption following plasma filtration in the glomerulus. Previous studies have tried to explore the potential mechanism of Cd^2+^-induced cytotoxicity of the renal proximal tubule cells, which will help find potential intoxication targets and explore novel renoprotective strategies [2,3]. However, primary cellular events and key effector molecules of Cd^2+^-induced cytotoxicity remain unclear.

Cd^2+^-disrupted intracellular calcium (Ca^2+^) homeostasis induced autophagy inhibition, promoted oxidative stress, and thereby mediated cell apoptosis [4,5,6,7,8]. Previous studies have shown that heavy metals (Cd^2+^, Pb^2+^) exposure induces subcellular Ca^2+^ redistribution by increasing Ca^2+^ release from the endoplasmic reticulum (ER) Ca^2+^ stored in primary rat proximal tubular cells [9,10,11]. In mouse renal tubular cells, our previous study showed that Cd^2+^ increased cytosolic Ca^2+^ levels by activating phospholipase C (PLC)-inositol 1, 4, 5-trisphosphate receptor (IP_3_R), which is one of the main Ca^2+^ release channels of ER [12]. In addition to the Ca^2+^ release channels, ER also has Ca^2+^ reuptake channel sarco/endoplasmic reticulum Ca^2+^-ATPase (SERCA) to maintain ER Ca^2+^ store level. In NIH 3T3 fibroblast cells, Cd^2+^ exposure decreased SERCA activity and inhibited the reuptake of Ca^2+^ from the cytoplasm back into the ER, which caused the disruption of ER Ca^2+^ homeostasis and cell apoptosis [13]. However, the roles of SERCA in the Cd^2+^-induced apoptosis of renal tubular cells have not been clarified.

SERCA activity and expression were significantly increased in human embryonic kidney cells HEK-293 with gain-of-function variants of the Ca^2+^ sensing receptor (CaSR) [14]. It suggests that SERCA might be a downstream target of CaSR. Moreover, NPS R-467-activated CaSR could restore Cd^2+^-disrupted intracellular Ca^2+^ homeostasis and protect against Cd^2+^-induced kidney apoptosis and injury [12,15]. Therefore, in this study, we aimed to identify the roles of SERCA in Cd^2+^-modulated Ca^2+^ homeostasis and Cd^2+^-induced ER stress and apoptosis in renal tubular cells and explore the underlying mechanism of Cd^2+^-targeted SERCA regulation. 

## 2. Results

### 2.1. NPS R-467 Protects against Cd^2+^-Induced Apoptosis Associated with Restoring Cytosolic Ca^2+^ Homeostasis 

Cd^2+^-induced nephrotoxicity originated from the damage of renal tubular cells. As shown in Figure 1A, Cd^2+^ treatment increased the expression of kidney injury biomarker, kidney injury molecule-1 (KIM-1) in mRTEC (Figure 1A). The results of the flow cytometry assay showed that Cd^2+^ treatment induced apoptosis of mRTEC cells in a dose-dependent manner (Appendix A). However, Cd^2+^ (5 μM)-induced apoptosis was effectively reversed by R-467 pretreatment (Figure 1B and Appendix A). The results of Fura-2/AM staining showed that Cd^2+^ treatment induced a transient increase of cytosolic Ca^2+^ levels in a few seconds, which is similar to TG (inhibitor of ER Ca^2+^ reuptake channel SERCA) treatment alone (Figure 1C,D), and was abrogated by the pretreatment of TG and 2-APB (inhibitor of ER Ca^2+^ release channel IP_3_R) (Figure 1E,F). Although NPS R-467 treatment alone did not change the cytosolic Ca^2+^ level, it could restore the Cd^2+^-increased cytosolic Ca^2+^ level in a few minutes (Figure 1G,H). These results showed that activation of CaSR by R-467 could protect against Cd^2+^-induced apoptosis associated with restoring cytosolic Ca^2+^ homeostasis, which might be mediated by ER. 

### 2.2. Cd^2+^-Induced Cytosolic Ca^2+^ Level Depends on ER Store

In order to investigate if part of the increased cytosolic Ca^2+^ level is dependent on extracellular Ca^2+^, we detected the change of cytosolic Ca^2+^ level after the short-term application of Cd^2+^ in Ca^2+^ free medium. The result of Flou-3/AM showed that the Ca^2+^ levels increased after short-term Cd^2+^ treatment (0.1 h) in both Ca^2+^-free HBSS and RMPI 1640 medium (Figure 2A). Then, the intracellular Ca^2+^ could return to resting level (0.5–4 h) (Figure 2A), which was further confirmed by the staining of Fura-2/AM after the cells were treated with Cd^2+^ for 4 h (Figure 2B). It suggests that Cd^2+^-induced elevation of cytosolic Ca^2+^ levels is not dependent on extracellular Ca^2+^ and could be restored in short-term Cd^2+^ treatment.

The ER is a main Ca^2+^ store in the cells and maintains intracellular Ca^2+^ homeostasis. As shown in Figure 2C, Mag-Fluo-4/AM is used to indicate ER Ca^2+^ content as it is specially co-localized or surrounded with ER-trackers in mRTEC (Figure 2C). Although short-term Cd^2+^ treatment (4 h) did not alter the ER Ca^2+^ content (Figure 2D), the long-term Cd^2+^ treatment (24 h) would reduce the ER Ca^2+^ content (Figure 2E). Moreover, SERCA agonist CDN1163 could restore the ER Ca^2+^ level disrupted by Cd^2+^ exposure (Figure 2E). Certainly, long-term Cd^2+^ treatment-reduced ER Ca^2+^ content might be due to Cd^2+^-induced cell apoptosis as TG treatment. The pretreatments of R-467 and CDN1163 could activate SERCA, the key Ca^2+^ reuptake channel of ER, to maintain Ca^2+^ homeostasis and inhibit Cd^2+^-induced apoptosis. Therefore, it needs further investigation in the following experiments.

### 2.3. SERCA Inhibition Induces ER Stress and Apoptosis

To further identify the role of SERCA in mRTEC cells, the effects of SERCA inhibitor TG and SERCA agonist CDN1163 on the ER stress, autophagy, and apoptosis were detected. Western blotting results showed that Cd^2+^ slightly increased the expression of ER stress biomarkers BiP and PDI, and the apoptosis biomarker cleaved caspase-3, although not as TG so significantly (Figure 3A–E and Appendix A). It is noted that CDN1163 could restore Cd^2+^-increased expression of BiP, PDI, and cleaved caspase-3 but not autophagy biomarker p62 (Figure 3A–E). Overexpression of SERCA2 (the main isoform of SERCA expressed in kidneys) could inhibit Cd^2+^-increased expression of PDI and cleaved caspase-3 and store cell viability (Figure 3F–H). These results suggest that the inhibitory effect of Cd^2+^ on SERCA contributes to ER stress and increases apoptosis of mRTEC cells but does not inhibit autophagy flux.

### 2.4. R-467 Reverses Cd^2+^-Inhibited SERCA2 and p-PLB Expression in Mice Kidney Cells and Tissues 

SERCA2 activity is regulated by the phosphorylation of PLB. To identify the regulation of Cd^2+^ on SERCA2, the expression of SERCA2 and p-PLB in mRTEC cells and kidney tissues was detected. As shown in Western blotting and immunofluorescence staining, Cd^2+^ significantly reduced the protein levels of SERCA2 and p-PLB in both mRTEC cells and renal tubules of kidney tissues (Figure 4A–D). Moreover, Cd^2+^-down-regulated SERCA2 and p-PLB expression were restored by R-467 (Figure 4A–D). These results confirmed that Cd^2+^ inhibited the expression and activity of SERCA2. The restoring effects of R-467 on expression and activity SERCA2, together with the cytosolic and ER Ca^2+^ levels, indicated that SERCA could act as a downstream target of CaSR.

### 2.5. Cd^2+^ Induces SERCA2 Protein Degradation through Proteasome Pathway 

To explore the mechanism of Cd^2+^-decreased expression of SERCA2, the mRNA level and protein stability of SERCA2 were detected. The RT-PCR results suggested that the SERCA2 mRNA level in mRTEC cells did not alter by Cd^2+^ exposure (Figure 5A,B). In addition, the results showed that Cd^2+^-induced SERCA2 degradation was inhibited after being treated with proteasome inhibitor MG132 but not lysosome inhibitor CQ (Figure 5C,D). These results indicated that Cd^2+^ exposure modulated SERCA2 expression in mRTEC cells by promoting the protein degradation pathway instead of inhibiting mRNA transcription. Cd^2+^-reduced SERCA2 proteins stability was further confirmed by a half-life assay. It showed that in the presence of the translation inhibitor, CHX, Cd^2+^ exposure reduced the half-life of SERCA2 (Figure 5E,F). Taken together, these data suggest that Cd^2+^ inhibited SERCA2 expression by stimulating the proteasomal degradation pathway.

## 3. Discussion

Cd^2+^ accumulated in the human kidney causes nephrotoxicity that promotes the development of CKD [16,17], as well as the occurrence and progression of renal cell carcinoma (RCC) [18,19]. The initiation of CKD and RCC originates from renal tubular cell injury and cancerization [18,20]. In previous studies, we showed that NPS R-467-activated CaSR could abrogate Cd^2+^-disrupted intracellular Ca^2+^ homeostasis and protect against long-term Cd^2+^ exposure-induced kidney apoptosis and injury [12,15]. In addition, it is supposed that ER Ca^2+^ homeostasis might play a key role in the Cd^2+^-induced apoptosis of renal tubular cells [12]. However, the key effectors targeted by Cd^2+^ have not been identified. Our results demonstrated that Cd^2+^ disrupted ER Ca^2+^ homeostasis by inhibiting SERCA2, which could be a potential target to prevent Cd^2+^-induced ER stress and cytotoxicity of renal tubular cells.

The ER is a sophisticated organelle controlling Ca^2+^ homeostasis by two Ca^2+^-releasing channels: IP_3_R and Ryanodine receptor (RyR), and a Ca^2+^ reflux channel SERCA. The balance between these three Ca^2+^ channels is crucial to maintain the dynamic balance of ER Ca^2+^ [21,22]. In addition, the Ca^2+^ channels on the plasma membrane, such as transient receptor potential vanilloid (TRPV) channels, could maintain ER Ca^2+^ homeostasis and protect against ER stress-induced apoptosis [23,24]. However, Cd^2+^ competites the TRPV5 and TRPV6 channels to block the Ca^2+^ uptake [25,26,27]. In proximal renal tubule epithelial cells, the Ca^2+^ release channel on the ER is mainly IP_3_R [28,29]. Through IP_3_R, Cd^2+^ induced Ca^2+^ release from the ER and caused Ca^2+^ redistribution, further resulting in primary rat proximal tubular cell apoptosis [9]. In mRTEC cells, our previous results confirmed that Cd^2+^ increased intracellular Ca^2+^ levels by activating the PLC-IP_3_R pathway [12]. Moreover, NPS R-467-activated CaSR could reverse Cd^2+^-disrupted cytosolic Ca^2+^ homeostasis and protect against Cd^2+^-induced kidney apoptosis and injury [12,15]. The present study identified that Cd^2+^-induced ER Ca^2+^ loss was effectively reversed by CaSR activator R-467 and SERCA agonist CDN1163. It suggests that Cd^2+^-disrupted ER Ca^2+^ homeostasis was modulated by the SERCA pump.

It suggests that SERCA activity plays an important role in Cd^2+^-induced cytotoxicity. Cd^2+^-decreased SERCA activity causes the disruption of ER Ca^2+^ homeostasis and apoptosis in NIH 3T3 fibroblast cells [13]. In the primary culture of rat cerebral cortical neurons, Cd^2+^ induced apoptosis by inhibiting Ca^2+^-ATPase activity [30]. In HepG2 cells, SERCA agonist CDN1163 can reverse Cd^2+^-increased triglyceride levels [31]. The SERCA pump activity is regulated by phospholamban (PLB). When binding to PLB, SERCA activity is suppressed; hence, the Ca^2+^ transport rate from the cytoplasm into ER decreases, which is reversed by the phosphorylation of PLB (p-PLB) [32]. Here, our results suggested that Cd^2+^ reduced the protein levels of p-PLB both in mouse renal tubular cells and kidney tissues. It suggests that Cd^2+^ induces ER stress, and apoptosis of renal tubular cells might be by inhibiting SERCA activity.

It is supposed that dysregulation of SERCA2 could cause kidney diseases. Reducing SERCA2-dependent ER stress could alleviate podocyte injury in diabetic nephropathy mice [33]. Upregulating SERCA2 expression causes ER Ca^2+^ depletion, which triggers apoptotic pathways in diabetic nephropathic mice [34]. Activation of AMP-activated protein kinase (AMPK) increased activation of SERCA to reduce cytosolic Ca^2+^ in vascular smooth muscle, which might benefit the treatment of vasodilation in human intrarenal arteries [35]. By attenuating SERCA2-dependent ER stress, Astragaloside IV prevented podocyte apoptosis in the progression of diabetic nephropathy [33]. The results here showed that Cd^2+^ reduced the protein level of SERCA2 and upregulated the expression of ER stress and apoptosis biomarkers, which was reversed by CaSR activator R-467 and SERCA agonist CDN1163, as well as overexpression of SERCA2. It suggests that Cd^2+^-inhibited expression of SERCA2 causes ER stress and apoptosis in renal tubular cells.

Our previous study suggested that Cd^2+^ inhibited autophagy flux by increasing Ca^2+^ released from ER through the IP_3_R channel [12]. With the Ca^2+^ channels IP_3_R and SERCA, the ER can transfer the cytosolic Ca^2+^ to the lysosomes to affect the lysosomal acid environment, an essential element for autophagy flux [36,37]. SERCA inhibitor Saikosaponin-D can inhibit the proliferation of autosomal dominant polycystic kidney disease (ADPKD) cells by promoting autophagy [38]. In liver HepG2 cells, the IP_3_R inhibitor 2-APB could restore the Cd^2+^-disrupted lysosomal acidic environment, which was also dysregulated by SERCA inhibitor TG [31]. Here, our result also confirms that TG treatment inhibits the autophagy flux as p62 accumulates in renal tubular cells. However, TG and Cd^2+^-induced p62 accumulation could not be reversed by R-467 and SERCA agonist CDN1163. It indicates that Cd^2+^ inhibits autophagy flux and might simultaneously increase IP_3_R-mediated Ca^2+^ release and suppress SERCA-dependent Ca^2+^ reflux.

Moreover, the lysosomal acidity inhibitor CQ could not inhibit Cd^2+^-induced SERCA2 degradation, which was obviously reversed by proteasomal inhibitor MG132. It suggests that Cd^2+^ induced SERCA2 degradation by promoting the proteasome pathway. The ubiquitin-proteasome system (UPS) is a main protein degradation process tightly related to ER stress [39]. ER stress inhibitor, 4-phenylbutyric acid (PBA), could prevent diabetic muscle wastage/atrophy in rats by reducing UPS activation [40] and reducing neuronal apoptosis of diabetes rats by restoring UPS activity [41]. Our results demonstrated that Cd^2+^ decreased SERCA2 stability by stimulating the proteasomal pathway in renal tubular cells.

The present study demonstrates that Cd^2+^ exposure induces apoptosis of renal tubular cells mRTEC due to disrupting ER Ca^2+^ homeostasis by suppressing SERCA2 expression and activity. SERCA2 agonist and overexpression of SERCA2 could abate Cd^2+^-induced ER stress and cell apoptosis. In addition, Cd^2+^ reduced SERCA2 protein stability by promoting proteasomal degradation. Therefore, SERCA2 might be a potential new target to protect against Cd^2+^-induced cytotoxicity and renal injury.

## 4. Materials and Methods

### 4.1. Reagents

Cadmium chloride, cycloheximide (CHX), and proteasome inhibitor MG132 were purchased from Sigma (Saint Louis, MO, USA). IP_3_R inhibitor 2-APB, SERCA inhibitor thapsigargin (TG), and SERCA agonist CDN1163 were purchased from Tocris Biosciences (Bristol, UK). Fura-2/AM, Fluo-3/AM, and Mag-Fluo-4/AM were purchased from Thermo Fisher Scientific (Shanghai, China); Calcimimetics NPS R-467 was purchased from NPS Pharmaceuticals (Salt Lake, UT, USA).

### 4.2. Animal Experiments and Cell Culture

All animal procedures complied with the guidelines approved by the Ethical Committee for the Use of Laboratory Animals of Jiangsu University (SYXK 2018-0053). As in our previous studies, 10 male ICR mice (8 weeks old, *n* = 5, around 25 g) were assigned to Cd^2+^ exposure group and control group (5 in each group), performed at least three times. CdCl_2_ was dissolved in distilled water and then added into Specific Pathogen Free (SPF) grade diets to final concentration of 1000 ppm Cd^2+^. Mice were continually fed on the Cd^2+^ contaminated diet for 28 days. To determine the effect of NPS R-467, the mice were fed on the 1000 ppm Cd^2+^ contaminated diet and tail vein injection with NPS R-467 (10 μM) in 20 μL PBS every day for 28 days. The mice were raised at 18~22 °C, with 50–60% humidity, natural daylighting, and tidy housing surrounding. All mice were euthanized by cervical dislocation at time point, and the kidney tissues were taken out for immunofluorescence staining. 

Mouse renal tubular epithelial cells (mRTECs) obtained from American Type Culture Collection (ATCC, Manassas, VA, USA) were cultured and maintained in RPMI 1640 medium (pH 7.4) supplemented with 10% fetal bovine serum (FBS) (Gibco, Thermo Fisher Scientific, Shanghai, China) and 1% penicillin/streptomycin in 5% CO_2_ incubator with 95% humidity. RPMI 1640 medium, FBS, penicillin, and streptomycin were purchased from Gibco (Thermo Fisher Scientific, Shanghai, China).

### 4.3. Measurement of Cell Viability

After being treated for 24 h, the cell viability was measured by the MTT (3-(4,5-dimethyl-2-thiazolyl)-2,5-diphenyl-2-H- tetrazolium bromide) assay (Sangon, Shanghai, China). In brief, cells were incubated in 100 μL MTT solution (0.5 mg/mL in RPMI 1640 medium) in 96-well plate for 4 h before the end of incubation. The supernatant was discarded, and 100 μL DMSO (Sangon, Shanghai, China) was added to dissolve the colored product (formazan). The absorbance was measured at 540 nm (690 nm as reference) using a Synergy H4 Hybrid Multi-Mode Microplate Reader (BioTek, Winooski, VT, USA).

### 4.4. Flow Cytometry

The apoptosis rate was determined using an Annexin V-FITC and propidium iodide (PI) double staining kit (Neobiosciences, Shenzhen, China) according to the manufacturer’s instructions. Briefly, after being treated with CdCl_2_ (1, 5, 10 μM) or CdCl_2_ (5 μM) + R-467 (1 μM) for 24 h, 1 × 10^6^ cells were re-suspended in binding buffer containing Annexin V-FITC (0.025%) and/or PI (20 μg/mL) and incubated in darkness at room temperature for 3 and 10 min, respectively. Then, the cells were immediately analyzed by a FACScan flow cytometer (Beckman Instruments, Fullerton, CA, USA).

### 4.5. RNA Isolation and RT-PCR Analysis

The cell total RNA was isolated using Trizol (Sangon, Shanghai, China). All RNA isolations were performed as directed by the manufacturer. Gel electrophoresis and ethidium bromide staining confirmed the purity and integrity of the samples. Quantification of RNA was based on spectrophotometric analysis at 260/280 nm. cDNA was made from 10 μg total RNA using an RNA-to-cDNA kit (Vazyme, Nanjing, China). Real-time PCRs were carried out by ABI7300 system using SYBR Green master mix (Vazyme, Nanjing, China). Gene-specific primers for Mouse-SERCA2-F: GGCATCTTTGGGGAGAACGA, Mouse SERCA2-R: CCATCACCTGTCATGGCTGT, Mouse-βActin-F: AAGCTGTGCTATGTTGCTCTA, Mouse-βActin-R: GTTTCATGGAT GCCACAGGA. The occurrence of primer-dimers and secondary products was inspected using melting curve analysis and agarose gel electrophoresis. Control amplification was done either without reverse transcriptase or without RNA. The relative expression ratio of a target gene was calculated according to their threshold cycle Ct values.

### 4.6. Western Blotting Analysis

The cells were treated with Cd^2+^ (5 μM) for 24 h. In some experiments, cells were either untreated or pre-treated with regulators, i.e., R-467 (1 μM), CQ (20 μM), and CDN1163 (10 μM) for 1 h, followed by treatment with Cd^2+^ (5 μM) for 24 h, or TG (0.1 μM) and MG132 (10 μM) for 6 h. The cells were lysed in RIPA buffer on ice for 30 min. After centrifugation at 13,000× *g* for 15 min at 4 °C, the supernatant was collected, and the total protein concentration was determined by BCA (Thermo Fisher Scientific, USA). The protein lysates containing 40 μg total cellular protein in RIPA buffer were subjected to electrophoresis on 8–12% polyacrylamide gels. The gels were then blotted onto PVDF membranes (Millipore, Billerica, MA, USA). Western blotting was conducted using mouse monoclonal antibody C/EBP homologous protein (CHOP) (Cell Signaling Technology, Shanghai, China) or rabbit monoclonal antibodies against phosphorylation phospholamban (p-PLB) (Abcam, Cambridge, MA, USA), Ubiquitin (Santa Cruz, Dallas, TX, USA), SERCA2, Binding immunoglobulin Protein (BiP), Protein Disulphide Isomerase (PDI), Inositol-requiring enzyme 1 (IRE-1α), endoplasmic reticulum oxidoreductase 1 (ERO1), p62, cleaved caspase-3 (Cell Signaling Technology, Shanghai, China) (1:1000), followed by incubation with horseradish peroxidase-conjugated goat anti-rabbit antibody (1:4000). Specific bands were visualized using ECL chemiluminescent reagent (Vazyme, Nanjing, China). The blots were then washed in PBST and re-probed with rabbit anti-actin (Cell Signaling Technology, Shanghai, China) (1:1000). 

### 4.7. Plasmid Transfection

SERCA2-GFP overexpression plasmid (VB190422-1024rhe), which contains full length of mouse SERCA2 expression cassette, was constructed by VectorBuilder (Guangzhou, China). Transfection was performed on 60–70% confluent mRTEC cells according to the manufacturer’s instruction of Lipofectamine 2000 (Thermo Fisher Scientific, Shanghai, China). Transfection mixture (per well in 6-well culture plates) consisted of 1.5 μL Lipofectamine 2000, 1.5 μg ultra-pure plasmid SERCA2-GFP, and 900 μL optimum reduced serum-free medium. Cells were washed once with optimum reduced serum-free medium, the transfection mixture was added and incubated for 6 h, and then the cell medium was replaced by fresh completed medium without antibiotics. After being cultured for 48 h, cells were treated with or without Cd^2+^ (5 μM) for 24 h before protein extraction.

### 4.8. Protein Half-Life of SERCA2

The cells were incubated with 5 μM Cd^2+^ for 24 h and cycloheximide (CHX, 10 μg/mL) for 0, 3, and 6 h. Total protein was extracted from those treated cells, and the amount of SERCA2 was estimated by Western blotting using anti-SERCA2 antibodies, which were used to determine the half-life of those proteins. Quantification of the relative protein levels of SERCA2 was performed using the software Image J 1.54c.

### 4.9. Immunofluorescence

To detect the expression of SERCA2 and p-PLB in mRTEC, the cells were treated with CdCl_2_ (5 μM) with or without R-467 (1 μM) for 24 h. The cells were fixed for 30 min in 4% Formaldehyde (Sangon, Shanghai, China). Then, the cells were permeabilized with 0.1% Triton X-100 (Sangon, Shanghai, China) in PBS for 20 min. Paraformaldehyde-fixed mice spleen sections were dewaxed, rehydrated in graded ethanol, and rinsed in PBS + 0.1% Tween 20 (PBST). The staining procedure involved incubation of the cells or tissue sections with 3% normal goat serum in PBST to reduce non-specific staining, followed by overnight incubation at 4 °C with rabbit polyclonal antibodies against SERCA2 (1:50, Cell Signaling Technology, Shanghai, China), p-PLB (1:50, Abcam, Shanghai, China) KIM-1 (1:200, Thermo Fisher Scientific, Shanghai, China) antibodies. Then, the cell or sides were incubated with Alexa Fluor 488 goat anti-rabbit IgG (1:200, Thermo Fisher Scientific, Shanghai, China) for 1 h at room temperature. The cells or tissues were stained by the DAPI (Thermo Fisher Scientific, Shanghai, China) for several minutes and were examined by Olympus IX73 microscopy (Tokyo, Japan). The cells or slides were washed for 3 × 15 min in PBST after each antiserum application. 

### 4.10. Measurement of Intracellular Ca^2+^ Level and ER Ca^2+^ Content 

To detect the transient effect of Cd^2+^ on intracellular Ca^2+^ levels in renal cells, mRTEC cells were seeded at a density of 5 × 10^3^ cells/well in 96-well plates. Next day, the cells were loaded with 1 μM Fura-2/AM in the RMPI 1640 medium (phenol red free) for 30 min at 37 °C in the dark. After dye loading, the cells were washed twice with the medium. Adding the medium with CdCl_2_ (5 μM), R-467 (1 μM) or CdCl_2_ (5 μM) + R-467 (1 μM), or pretreated with TG (0.1 μM), 2-APB (50 μM). Then, the fluorescent intensity was recorded with excitation at 348 nm and emission at 380 nm using a Synergy H4 Hybrid Multi-Mode Microplate Reader (BioTek, Winooski, VT, USA).

For measurement of the short-term effect of Cd^2+^ on cytosolic Ca^2+^ levels, the cells were treated with CdCl_2_ (5 μM) for 0–4 h. For long-term effect of ER Ca^2+^ content, the cells were treated with CdCl_2_ (5 μM) for 24 h, or pre-treated with regulators, i.e., R-467 (1 μM), CDN1163 (10 μM) for 1 h, followed by treatment with Cd^2+^ (5 μM) for 24 h, or TG (0.1 μM) for 6 h. Then, the cells were washed twice with RMPI 1640 medium (phenol red free), followed by loading with 1 μM Mag-Fluo-4/AM (for ER Ca^2+^ content, Thermo Fisher Scientific, Shanghai, China) in the RMPI 1640 medium (phenol red free) for 30 min at 37 °C in the dark. After dye loading, the cells were washed twice with the medium. Then, the fluorescent intensity was recorded with excitation at 488 nm and emission at 535 nm using a Synergy H4 Hybrid Multi-Mode Microplate Reader (BioTek, Winooski, VT, USA). Moreover, cells were loaded with ER-specific dye ER-tracker red (Beyotime, Shanghai, China) and/or Mag-Fluo-4-AM to indicate that Mag-Fluo-4 was selectively labeled on ER. The co-localization of Mag-Fluo-4 with ER-tracker red was observed by Leica TCS SP5 confocal microscope (Heidelberg, Germany). 

### 4.11. Statistical Analysis

Chemical treatments were performed in triplicate in each experiment, and every experiment was repeated at least three times. All data are represented as means ± standard deviation (SD). Statistical significance was assessed with Student’s *t*-test or one-way analysis of variance (ANOVA) followed by Duncan’s multiple range test by IBM SPSS Statistics 27.0.1 software as described in figure legends. Groups were considered significantly different if * *p* < 0.05.

## Figures and Tables

**Figure 1 ijms-24-05979-f001:**
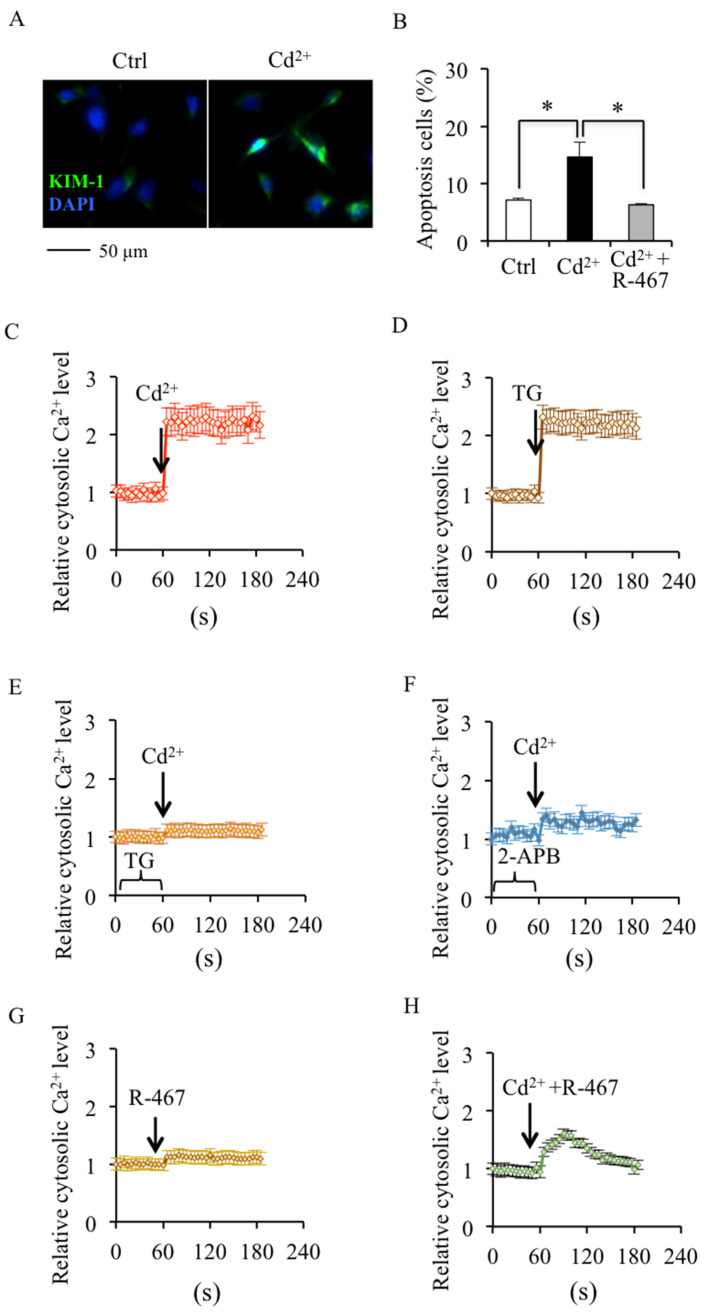
Effects of NPS R-467 on Cd^2+^-induced apoptosis and dysregulated transient cytosolic Ca^2+^ homeostasis. (**A**) Cd^2+^ induced the expression of kidney injury biomarker KIM-1 in mRTEC. KIM-1: Green, DAPI: Blue. (**B**) Effects of R-467 on Cd^2+^-induced apoptosis of mRTEC cells. After being treated with CdCl_2_ (5 μM) or CdCl_2_ (5 μM)+R-467 (1 μM) for 24 h, the apoptosis of mRTEC cells was determined by flow cytometry staining by Annexin V-FITC/PI. The apoptosis percentage was calculated. (**C**–**H**) Transient effects of Cd^2+^ and R-467 on cytosolic Ca^2+^ levels in mRTEC cells. After treated with CdCl_2_ (5 μM) or CdCl_2_ (5 μM)+R-467 (1 μM) or pretreated with TG (0.1 μM) and 2-APB (50 μM), the cytosolic Ca^2+^ level changes in mRTEC cells were determined by Fura-2/AM immediately. 2-APB: inhibitor of ER Ca^2+^ release channel IP_3_R, R-467: Calcimimetics, activator of Calcium sensing receptor, TG: inhibitor of SERCA. Results are presented as mean ± SD (*n* = 4 well cells/group). * Statistical significance between control and treatments, or between treatments, *p* < 0.05, using one-way ANOVA followed by Duncan’s multiple range test.

**Figure 2 ijms-24-05979-f002:**
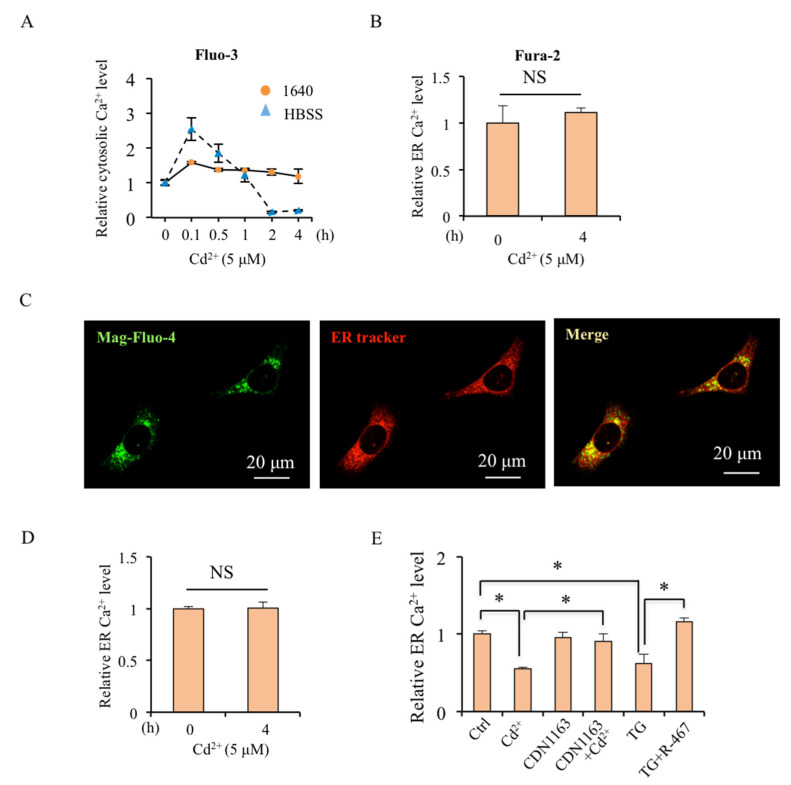
Effects of ER Ca^2+^ channels in Cd^2+^-dysregulated ER Ca^2+^ homeostasis. (**A**) Short-term effects of Cd^2+^ on cytosolic Ca^2+^ content in mRTEC cells. Cells were treated with CdCl_2_ (5 μM) for 0–4 h, and the cytosolic Ca^2+^ levels in mRTEC cells were determined by Fluo-3/AM in Ca^2+^ containing and Ca^2+^-free medium, respectively. (**B**) Steady cytosolic Ca^2+^ levels in mRTEC cells were determined by Fura-2/AM after being treated with CdCl_2_ (5 μM) for 4 h. (**C**) Co-localization of ER Ca^2+^ content indicator Mag-Fluo-4/AM staining (Green) with ER tracker (Red). (**D**) Effects of short-term Cd^2+^ treatment on ER Ca^2+^ content. After being treated with Cd^2+^ (5 μM) for 4 h, the ER Ca^2+^ content in mRTEC cells was determined by Mag-Fluo-4/AM. (**E**) Effects of SERCA modulators on Cd^2+^-regulated ER Ca^2+^ content in mRTEC cells. After being treated with Cd^2+^ (5 μM) for 24 h or pretreated with CDN1163 (10 μM) or R-467 (1 μM) for 1 h, followed with TG (0.1 μM) for 6 h. The ER Ca^2+^ content in mRTEC cells was determined by Mag-Fluo-4/AM. Results are presented as mean ± SD (*n* = 4 well cells/group). * Statistical significance difference between control and Cd^2+^ treatment or between treatments. NS—No significant difference. *p* < 0.05, using one-way ANOVA followed by Duncan’s multiple range test. CDN1163: agonist of ER Ca^2+^ reuptake channel SERCA, TG: inhibitor of SERCA.

**Figure 3 ijms-24-05979-f003:**
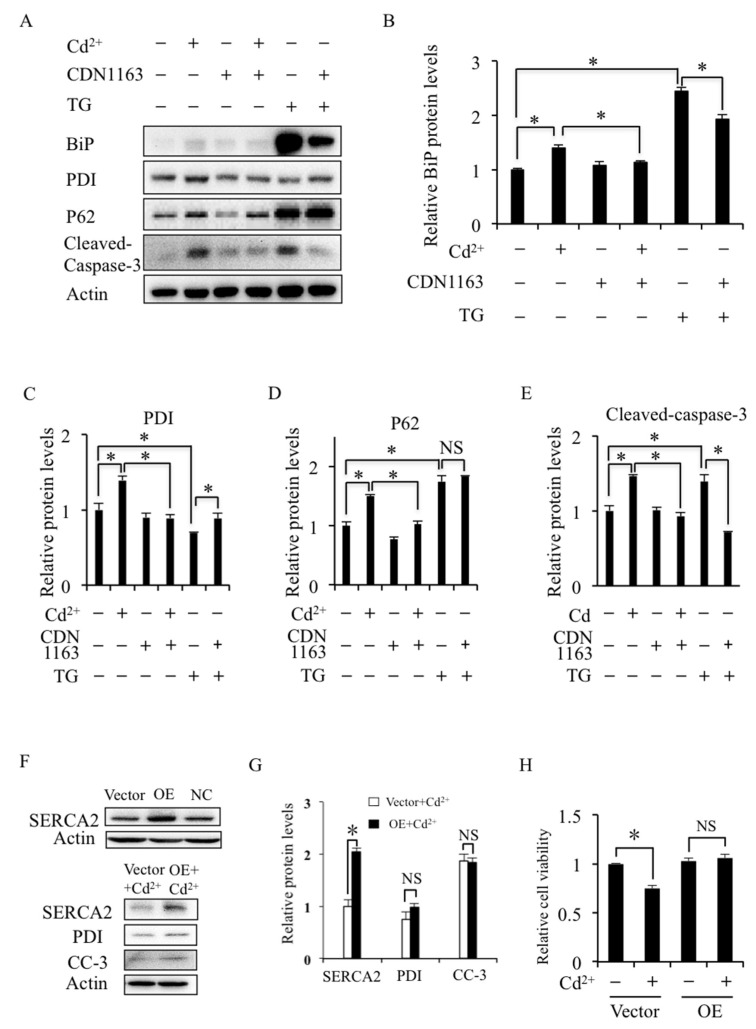
Effects of SERCA modulators on ER stress, autophagy, and apoptosis. (**A**) Effect of Cd^2+^, CDN1163, and TG on the expression of ER stress biomarkers BiP and PDI, autophagy biomarker p62, and apoptosis biomarker cleaved caspase-3 (CC-3) were detected by Western blotting. (**B**–**E**) Quantification of the relative protein levels was performed using the software Image J 1.54c. (**F**,**G**) Effect of Cd^2+^ on the expression of PDI and CC-3 detected by Western blotting and quantification by Image J. NC—negative control. OE: SERCA2 overexpression. (**H**) MTT detection of cell viability in Cd^2+^-treated SERCA2 overexpressed mRTEC cells. Results are presented as mean ± SD (*n* = 4 well cells/group). * indicates statistical significance between control and Cd^2+^ treatment or between treatments. NS means no significant difference. *p* < 0.05, using one-way ANOVA followed by Duncan’s multiple range test.

**Figure 4 ijms-24-05979-f004:**
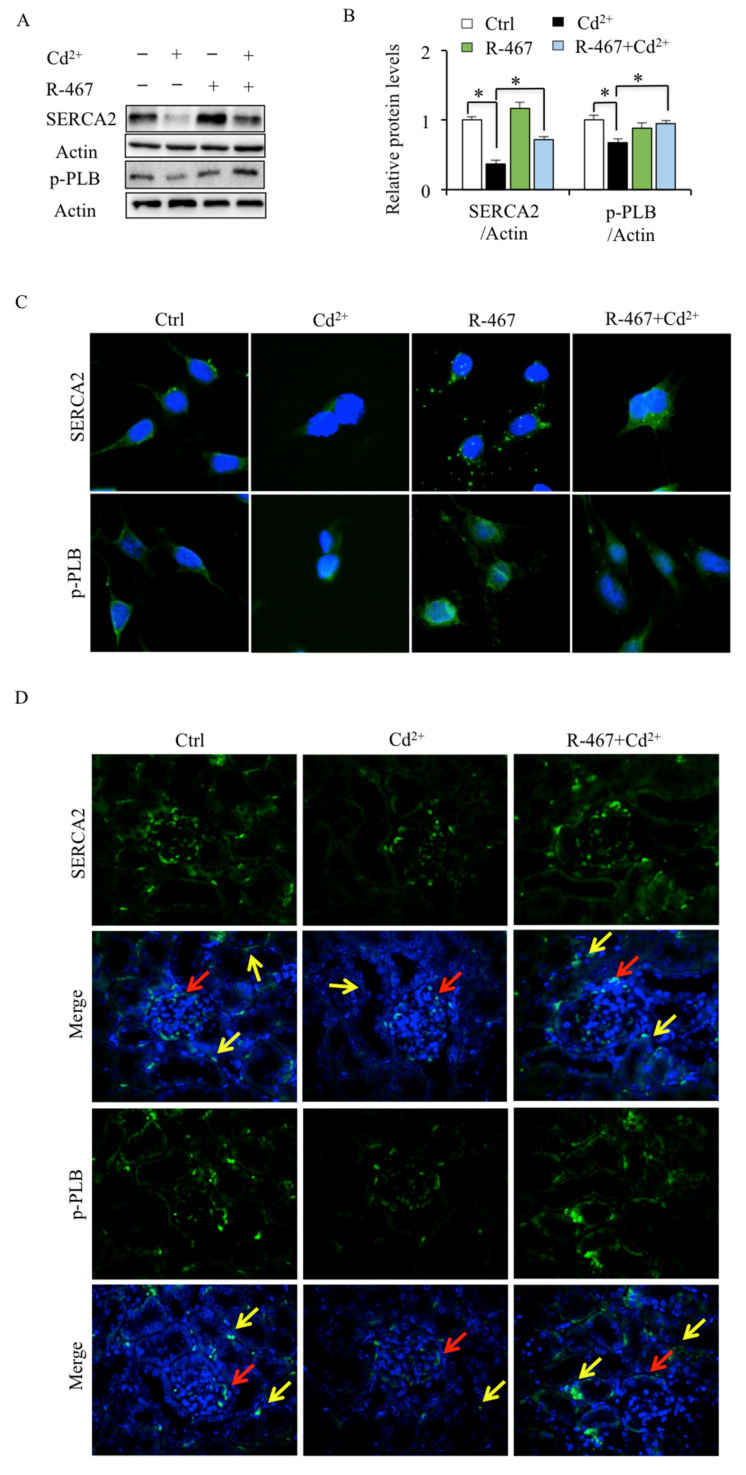
Effects of R-467 on Cd^2+^-regulated SERCA2 and p-PLB expression in mice kidney cells and tissues. (**A**) Effects of R-467 on Cd^2+^-regulated SERCA2 and p-PLB expression in mRTEC cells were detected by Western blotting. (**B**) Quantification of the relative protein levels was performed using the software Image J 1.54c. Results are presented as mean ± SD (*n* = 4 well cells/group). * Statistical significance between control and Cd^2+^ treatment or between treatments. *p* < 0.05, using one-way ANOVA followed by Duncan’s multiple range test. Effects of R-467 on Cd^2+^-regulated SERCA2 and p-PLB expression in mRTEC cells (**C**) and mice kidney tissues (**D**) were detected by immunohistochemistry. After being treated by Cd^2+^ for 24 h, with or without R-467 pretreatment, cells were fixed (*n* = 4 well cells/group). After being exposed to Cd^2+^ for 28 days, mice kidney tissues were collected (*n* = 5 mice/group). The protein levels of SERCA2 and p-PLB in cells and tissues were investigated by immunofluorescence. SERCA2 (green), p-PLB (green), DAPI (blue). The red arrow refers to the glomerulus, and the yellow arrow refers to the renal proximal tubule.

**Figure 5 ijms-24-05979-f005:**
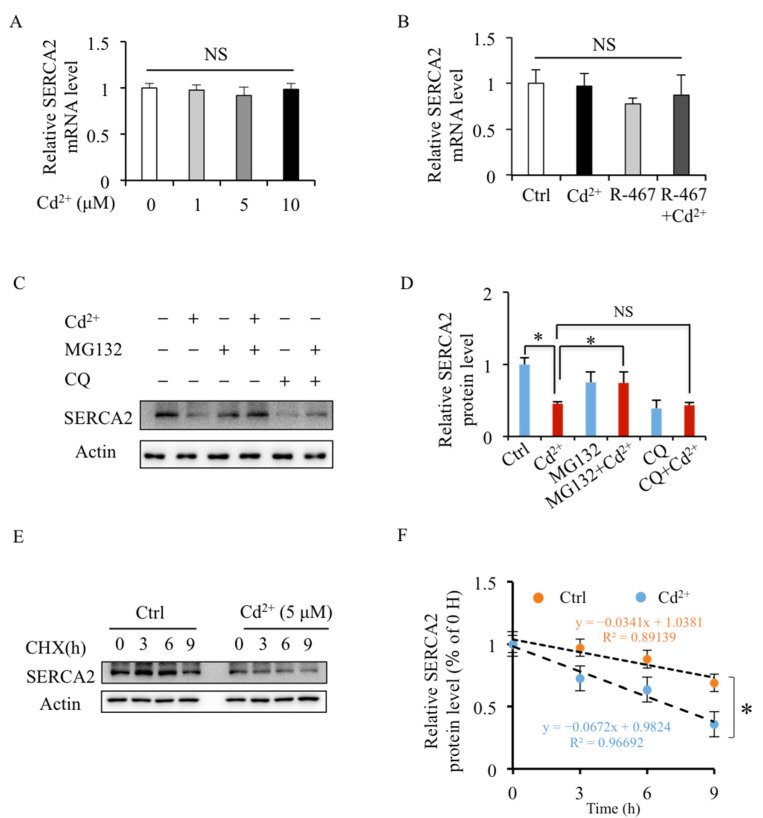
Effects of Cd^2+^ on SERCA2 stability in mRTEC cells. (**A**,**B**) After being treated with Cd^2+^ (0–10 μM) for 24 h or treated with Cd^2+^ (5 μM) or Cd (5 μM) + R-467 (1 μM) for 24 h, mRTEC cells were collected for RNA extraction, and the SERCA2 mRNA levels detected by RT-PCR. (**C**) Effects of MG132 and CQ on Cd^2+^-induced degradation of SERCA2. The mRTEC cells were pretreated with CQ (20 μM) for 1 h, followed by treatment with Cd^2+^ (5 μM) for 24 h, or Cd^2+^ (5 μM) for 18 h + MG132 (10 μM) for 6 h, and then the proteins were extracted for Western blotting analysis. (**E**) Effects of Cd^2+^ on half-life of SERCA2 protein. Cells were treated with Cycloheximide (CHX) (10 μg/mL) for 3 and 6 h or pretreated with Cd^2+^ (5 μM) for 24 h, and the protein was extracted for Western blotting analysis. (**D**,**F**) Quantification of the relative protein levels of SERCA2 was performed using the software Image J 1.54c. * indicates statistical significance between control and Cd^2+^ treatment or between treatments. NS: no significant difference. (*n* = 4 well cells/group) *p* < 0.05, using one-way ANOVA followed by Duncan’s multiple range test (**A**,**B**,**D**) and Student’s *t*-test (**F**).

## Data Availability

All the data supporting the conclusions is included within the manuscript and is available on request from the corresponding authors.

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
