# Peer review of "Cadmium Disrupted ER Ca^2+^ Homeostasis by Inhibiting SERCA2 Expression and Activity to Induce Apoptosis in Renal Proximal Tubular Cells"

_ijms, 2023, doi:10.3390/ijms24065979_

Round 1
Reviewer 1 Report (Previous Reviewer 1)
The manuscript by Li K, et al. improved very much. Please specify the soft used for the statistical analysis.
Author Response
Thanks for your comment that is valuable and helpful for improving our article.
Response: We added information in 4.11 Statistical analysis, IBM SPSS Statistics 27.0.1 software.
Reviewer 2 Report (New Reviewer)
The manuscript is a continuation of previous research on heavy metals adverse effects on kidneys. Authors have previously revealed that Cd2+ causes overload of the ER with Ca2+ which ultimately leads to cytotoxicity and apoptosis. In current manuscript, authors have focused on the protective effect of CaSR overexpression and calcimimetic NPS R-467 on renal tubular cells. The presented research suggests that sarco/endoplasmic reticulum Ca2+-ATPase, type 2, (SERCA2) is responsible for restoration of calcium homeostasis under Cd2+ treatment. Also, authors identified proteosomal degradation as main regulator of SERCA2 protein stability. So, the presented research not only has fundamental significance but also identifies SERCA2 as prospective target protein for kidney protection against cadmium exposure.
The research is well-designed, and conclusions are based on the experimental evidence. Nevertheless, some issues arise concern and should be resolved:
- Authors use Cd throughout the manuscript and Cd2+ in the Discussion section. My understanding is that Cd means cadmium, as metal. The metal itself does not cause any adverse effects as it is not soluble in water. So, most of the time authors mean ions of cadmium, soluble in water. If it is so, then the correction of Cd to Cd2+ is needed.
- Authors use Fura2/AM for measurement of cytosolic Ca2+ levels. The dye is ratiometric and is considered the best for calcium imaging. However, occasionally authors also use Fluo-3 which only allows to relatively estimate the Ca2+ change. What is the rationale behind using Fluo-3?
- Authors use Mag-Fluo-4 as indicator of ER calcium. But the dye itself is not specifically designed for that. More explanation on that is needed. The dye detects Mg2+ ions and is low affinity Ca2+ indicator. So why is it so handy in case of presented experiment?
The manuscript is relatively well written and clear, though minor corrections are needed. Also, the English editing would be highly recommended. Some sentences are hard to understand.
The minor corrections are as follows:
In Title:
Lines 2-4 – the nouns need some verb in the title. As for now, the title is just a phrase.
In Abstract:
Line 23 – “activity keeper” is not an established term in the field. Authors probably mean upstream regulator.
In Results:
Figure 1 – the arrow signifies the addition of Cd2+ but on Fig. 1 E and F authors need to mark the period of TG and 2-APB treatment, too. Usually, the horizontal line is added. Also, the calcium measurements represent mean ± SD, but the SD is not visible.
Line 86 – apoptosis was, not were.
Figure 4, C and D – authors should describe more precisely which is which.
Line 187 – there is typo, correct to SERCA.
Figure 5, C and E – C is describing as Western blotting analysis of SERCA2 half-life and E – as CQ and MG132 treatment. But actual image is quite the opposite.
In Materials and Methods:
Line 299 – BAPTA/AM is indicated as one of the reagents used in the study. Yet there is no result obtained with BAPTA. Authors should not indicate chemicals that have not been used in the study.
Line 309 – correct CdCl2 to CdCl2.
Line 334 – correct 1x106 to 1x10^6.
Line 420 – correct to Ca2+.
Line 431 – the mean ± SEM is not really mean ± SD. The description to Figures suggests the SD appears on the pictures. So, which one. Also, what does n=4 refer to? Do you mean repeats of number of cells/animals in the analysis?

Author Response
Responses to reviewer comments 2
These comments are all valuable and helpful for improving our article. We have tried our best to revise our manuscript according to the comments. We have revised the manuscript with track changes. Point-by-point responses to the reviewers are listed below.
In general, we hope our amended to conform to your comments. Although it is impossible to achieve perfect, but look in so much effort and workload, I sincerely hope you agree to accept.
The manuscript is a continuation of previous research on heavy metals adverse effects on kidneys. Authors have previously revealed that Cd2+ causes overload of the ER with Ca2+ which ultimately leads to cytotoxicity and apoptosis. In current manuscript, authors have focused on the protective effect of CaSR overexpression and calcimimetic NPS R-467 on renal tubular cells. The presented research suggests that sarco/endoplasmic reticulum Ca2+-ATPase, type 2, (SERCA2) is responsible for restoration of calcium homeostasis under Cd2+ treatment. Also, authors identified proteosomal degradation as main regulator of SERCA2 protein stability. So, the presented research not only has fundamental significance but also identifies SERCA2 as prospective target protein for kidney protection against cadmium exposure.
The research is well-designed, and conclusions are based on the experimental evidence. Nevertheless, some issues arise concern and should be resolved:
- Authors use Cd throughout the manuscript and Cd2+in the Discussion section. My understanding is that Cd means cadmium, as metal. The metal itself does not cause any adverse effects as it is not soluble in water. So, most of the time authors mean ions of cadmium, soluble in water. If it is so, then the correction of Cd to Cd2+is needed.
Response: We have correct to Cd2+ in the full manuscript.
- Authors use Fura2/AM for measurement of cytosolic Ca2+levels. The dye is ratiometric and is considered the best for calcium imaging. However, occasionally authors also use Fluo-3 which only allows to relatively estimate the Ca2+change. What is the rationale behind using Fluo-3?
Response: The higher Kd and longer excitation wavelength of fluo-3 can have significant advantages over fura-2. Although neutrophils and platelets are used as examples, these observations will be applicable to other cell types. The Kd of fluo-3 for binding Ca2+ at 37 degrees C was measured and found to be 864 nM; the previously published value was 400 nM at 22 degrees C. The Kd of fluo-3, like that of fura-2, is therefore very temperature-dependent. Protocols for loading cells, and preventing leakage of fluo-3, are described; probenecid, known to inhibit fura-2 leakage from cells, was found to be essential to get good fluo-3 signals from platelets. Calibration of fluo-3 fluorescence signals to [Ca2+] and methods for obtaining maximum and minimum fluorescence signals are described; these methods differ from those used with fura-2. Agonist-stimulated responses of fluo-3-loaded neutrophils and platelets are shown, and the calculated cytosolic [Ca2+] is comparable with that previously obtained with fura-2. Responses of cells in the presence of plasma are also shown; such measurements, unobtainable with quin2, fura-2 or indo-1, are possible with fluo-3, owing to its longer excitation wavelengths. Co-loading of cells with bis-(o-aminophenoxy)ethane-NNN'N'-tetra-acetic acid and fluo-3 is included as an example of how cytosolic [Ca2+] can be buffered and manipulated. Many of these observations will be of value when using fluo-3 (or other Ca2+-indicator dyes) in most cell types.
Moreover, we also verified the effects of short-term Cd2+ exposure (4 h) on cytosolic Ca2+ level by fura-2 (Figure 2B).
Reference:
Merritt JE, McCarthy SA, Davies MP, Moores KE. Use of fluo-3 to measure cytosolic Ca2+ in platelets and neutrophils. Loading cells with the dye, calibration of traces, measurements in the presence of plasma, and buffering of cytosolic Ca2+. Biochem J. 1990 Jul 15;269(2):513-9. doi: 10.1042/bj2690513.
Bito V, Sipido KR, Macquaide N. Basic methods for monitoring intracellular Ca2+ in cardiac myocytes using Fluo-3. Cold Spring Harb Protoc. 2015 Apr 1;2015(4):392-7. doi: 10.1101/pdb.prot076950.
- Authors use Mag-Fluo-4 as indicator of ER calcium. But the dye itself is not specifically designed for that. More explanation on that is needed. The dye detects Mg2+ions and is low affinity Ca2+indicator. So why is it so handy in case of presented experiment?
Response:
Synthetic Ca2+ indicators are widely used to report changes in free [Ca2+], usually in the cytosol but also within organelles. Mag-Fluo-4, loaded into the endoplasmic reticulum (ER) by incubating cells with Mag-Fluo-4 AM, has been used to measure changes in free [Ca2+] within the ER, where the free [Ca2+] is estimated to be between 100 μM and 1 mM. Many results are consistent with Mag-Fluo-4 reliably reporting changes in free [Ca2+] within the ER, but the results are difficult to reconcile with the affinity of Mag-Fluo-4 for Ca2+ measured in vitro (KDCa ∼22 μM). Using an antibody to quench the fluorescence of indicator that leaked from the ER, we established that the affinity of Mag-Fluo-4 within the ER is much lower (KDCa ∼1 mM) than that measured in vitro. We show that partially de-esterified Mag-Fluo-4 has reduced affinity for Ca2+, suggesting that incomplete de-esterification of Mag-Fluo-4 AM within the ER provides indicators with affinities for Ca2+ that are both appropriate for the ER lumen and capable of reporting a wide range of free [Ca2+].
In our present study, we aim to identify the role of Cd2+ on SERCA, which is the Ca2+ reuptake channel on ER. The change of influx of free [Ca2+] into the ER could be indicated by Mag-fluo-4/AM.
Reference:
Ana M Rossi, Colin W Taylor, Reliable measurement of free Ca2+ concentrations in the ER lumen using Mag-Fluo-4. Cell Calcium. 2020 May;87:102188. doi: 10.1016/j.ceca.2020.102188.
- The manuscript is relatively well written and clear, though minor corrections are needed. Also, the English editing would be highly recommended. Some sentences are hard to understand.
The minor corrections are as follows:
In Title:
Lines 2-4 – the nouns need some verb in the title. As for now, the title is just a phrase.
Response: we revised the title into: Cadmium disrupted ER Ca2+ homeostasis by inhibiting SERCA2 expression and activity to induce apoptosis in renal prox-imal tubular cells
- In Abstract:
Line 23 – “activity keeper” is not an established term in the field. Authors probably mean upstream regulator.
Response: we revised “keeper” to “regulator”.
- In Results:
Figure 1 – the arrow signifies the addition of Cd2+ but on Fig. 1 E and F authors need to mark the period of TG and 2-APB treatment, too. Usually, the horizontal line is added. Also, the calcium measurements represent mean ± SD, but the SD is not visible.
Response: We marked he period of TG and 2-APB treatment before 60 seconds. And we added the SD in figure 1.
- Line 86 – apoptosis was, not were.
Response: We have corrected it.
- Figure 4, C and D – authors should describe more precisely which is which.
Response: We indicated Figure 4C is from cells and 4D is from tissues.
- Line 187 – there is typo, correct to SERCA.
Response: We have corrected it.
- Figure 5, C and E – C is describing as Western blotting analysis of SERCA2 half-life and E – as CQ and MG132 treatment. But actual image is quite the opposite.
Response: We have corrected the figure legend of figure C and E.
- In Materials and Methods:
Line 299 – BAPTA/AM is indicated as one of the reagents used in the study. Yet there is no result obtained with BAPTA. Authors should not indicate chemicals that have not been used in the study.
Response: We have deleted it, as well as some parts.
- Line 309 – correct CdCl2 to CdCl2. Line 334 – correct 1x106 to 1x10^6. Line 420 – correct to Ca2+.
Response: We have corrected these superscript and subscript.
- Line 431 – the mean ± SEM is not really mean ± SD. The description to Figures suggests the SD appears on the pictures. So, which one. Also, what does n=4 refer to? Do you mean repeats of number of cells/animals in the analysis?
Response: It’s SD, and we have added n=4 well cells/group, n=5 mice/group.
This manuscript is a resubmission of an earlier submission. The following is a list of the peer review reports and author responses from that submission.
Round 1
Reviewer 1 Report
Gu J, et al. investigated the role of SERCA2 expression for cadmium-induced renal injury. The investigation was well designed. But the statistical analyses are completely wrong.
In Figure 1, R467 decreased cadmium-induced apoptosis. The changes of the percentages of apoptotic cells are quite small. Are there statistically significant.
In Figure 1and 4, the authors used Student’s t-test to compare. Since there are four groups, t-test cannot be used. ANOVA and multiple comparison are required. In other figures, the authors mentioned that they used t-test or ANOVA and multiple comparison. Student’s t-test can be used only in Figure 5F. The statistical analyses must be changed.
The authors used mRTECs. However, the authors did not show any evidence that the cells have the characteristics of proximal tubules in the present study.
Author Response
- Gu J, et al. investigated the role of SERCA2 expression for cadmium-induced renal injury. The investigation was well designed. But the statistical analyses are completely wrong.
Response: Thanks for your comments. We have clarified the statistical analyses in methods and figure legends.
- In Figure 1, R467 decreased cadmium-induced apoptosis. The changes of the percentages of apoptotic cells are quite small. Are there statistically significant.
Response: We did a statistical analysis as shown in figure 1 B. It suggested that there are significant difference between control and Cd treatment, Cd treatment vs Cd+R467.
- In Figure 1 and 4, the authors used Student’s t-test to compare. Since there are four groups, t-test cannot be used. ANOVA and multiple comparison are required. In other figures, the authors mentioned that they used t-test or ANOVA and multiple comparison. Student’s t-test can be used only in Figure 5F. The statistical analyses must be changed.
Response: We have clarified the statistical analyses in figure legends.
- The authors used mRTECs. However, the authors did not show any evidence that the cells have the characteristics of proximal tubules in the present study.
Response: Cd-induced nephrotoxicity originated from the damage of renal tubular cells. Here, we added a result that Cd induced the injury of mRTEC. As shown in figure 1A, Cd treatment increased the expression of kidney injury biomarker, kidney injury molecule-1 (KIM-1). (Result 2.1)
Reference:
Gu J, Ren Z, Zhao J, Peprah FA, Xie Y, Cheng D, Wang Y, Liu H, Chu Wong CK, Zhou Y, Shi H. Calcimimetic compound NPS R-467 protects against chronic cadmium-induced mouse kidney injury by restoring autophagy process. Ecotoxicol Environ Saf. 2020 Feb;189:110052.
Han WK, Bailly V, Abichandani R, Thadhani R, Bonventre JV. Kidney Injury Molecule-1 (KIM-1): a novel biomarker for human renal proximal tubule injury. Kidney Int. 2002 Jul;62(1):237-44
Prozialeck WC, Vaidya VS, Liu J, Waalkes MP, Edwards JR, Lamar PC, Bernard AM, Dumont X, Bonventre JV. Kidney injury molecule-1 is an early biomarker of cadmium nephrotoxicity. Kidney Int. 2007 Oct;72(8):985-93
Reviewer 2 Report
Major comments :
1) measurements of restig cytosolic concentration of calcium ions have been done with single emmisision and single absorption fluorescent probe Fluo3. This must be done with at least Fura-2 ratiometric probe and with calibration in order to avoid artefacts and to compare different cell population.
2) Effect of Cd ions on intracellular calcium concentration. The authors shold present some recording in conditions of zero calcium in extracellular bath in order to show the independency of responses to calcium entries.
3) The authors should show the effect of Tg pretreatment on Cd-dependent calcium transients
4) the authors should show the accumulation of the low affinity Ca probe Mg-Fluo4 in ER, by co-staining with a ER fluorescent probe
5) How do the authors explain the relationship between Cd-dependent calcium responses and the CaSR ?
6) I found some mistakes in legends of figures, for instance in figure 2.
7) Effects of TG, SERCA inhibitor, on ER Ca content and apoptosis are already well documented. This are not very original results and this does not particularly strenghten the conclusions on the effect of Cd ions on the ER.
8) I am not convinced by the "representing" wester-blot shown in figures (suplementary is better): this blot does not show convincing effects on BiP, PDI and P62. The only effect clearly visible is the effect of Cd on cleaved caspase 3 in coherence with graphs presenting mean values. There is also an expected high effect of Tg on BiP, P62 and cleaved caspases 3
9) the authors should compare the effects of Cd on SERCA expression with the expression of the ER calcium-release channel IP3R.
10) the authors should at least discuss known effects of Cd as a bloker of Calcium channels at the plasma membrane.
11) editing and english changes are required
Author Response
1) measurements of resting cytosolic concentration of calcium ions have been done with single emission and single absorption fluorescent probe Fluo3. This must be done with at least Fura-2 ratio metric probe and with calibration in order to avoid arte facts and to compare different cell population.
Response: Fura-2 is usually used as a probe to detect the transient cytosolic Ca2+ level changes as showed in figure 1C-F. Fluo-3 could be applied to measure the cytosolic Ca2+ level at relative steady state (figure 2C-F, G).
2) Effect of Cd ions on intracellular calcium concentration. The authors should present some recording in conditions of zero calcium in extracellular bath in order to show the independency of responses to calcium entries.
Response: As we known, the intracellular and extracellular Ca2+ levels of cells are in a dynamic equilibrium. If the cells in a zero calcium in extracellular bath condition, Ca2+ will efflux from the cells (intracellular Ca2+ store and cytosolic) to the extracellular to adapt the stress condition. For a long-term bath in the zero calcium medium, the cells will be detached and round. In these statues, it could not identify the role of treatments on the changes of cytosolic Ca2+ levels.
3) The authors should show the effect of Tg pretreatment on Cd-dependent calcium transients
Response: We added the result of transient cytosolic Ca2+ level changes of TG treatment (Figure 1 F).
4) the authors should show the accumulation of the low affinity Ca probe Mg-Fluo4 in ER, by co-staining with a ER fluorescent probe.
Response: We detected the co-localization of ER Ca2+ content indicator Mag-Fluo-4/AM staining (Green) with ER tracker (Red). (Figure 2A).
5) How do the authors explain the relationship between Cd-dependent calcium responses and the CaSR?
Response:
Cd induces Ca2+ release from ER store, associated with ER stress through cation-sensing receptor (CSR) mediated phospholipase C (PLC)-inositol 1, 4, 5-trisphosphate (IP3) signaling pathway (Biagioli et al., 2008; Faurskov & Bjerregaard, 2002). In addition, although due to CaSR agonist neomycin and Gd3+ (Gadolinium ion) could not stimulate CSR, suggesting CaSR is different from CSR, both receptors mediate activation of PLC-IP3 pathway and intracellular Ca2+ level (Faurskov & Bjerregaard, 2002). Our previous study demonstrated that activation of CaSR by calcimimetic R-467 reduced Cd-evoked intracellular Ca2+ elevation, followed ROS generation and apoptosis, and R-467 also restored autophagic flux and increased cell proliferation by switching Cd-activated calcium-p38 MAPK to R-467 activated PLC-ERK pathway (Gu et al., 2018).
SERCA activity and expression were significantly increased in human embryonic kidney cells HEK-293 with gain-of-function variants of the Ca2+ sensing receptor (CaSR) (Ranieri et al., 2013). It suggests that SERCA acts as a downstream target of CaSR. In the present study, our results showed that R-467 could restore the ER and cytosolic Ca2+ levels by regulating SERCA. (Result 2.4).
Reference:
Biagioli, M. et al. Endoplasmic reticulum stress and alteration in calcium homeostasis are involved in cadmium-induced apoptosis. Cell Calcium. 43, 184-195 (2008).
Faurskov, B. & Bjerregaard, H.F. Evidence for cadmium mobilization of intracellular calcium through a divalent cation receptor in renal distal epithelial A6 cells. Pflügers Arch. 445, 40 (2002).
Gu J, Dai S, Liu Y, Liu H, Zhang Y, Ji X, Yu F, Zhou Y, Chen L, Tse WKF, Wong CKC, Chen B, Shi H. Activation of Ca2+-sensing receptor as a protective pathway to reduce Cadmium-induced cytotoxicity in renal proximal tubular cells. Sci Rep. 2018 Jan 18;8(1):1092.
Ranieri M, Tamma G, Di Mise A, Vezzoli G, Soldati L, Svelto M, Valenti G. Excessive signal transduction of gain-of-function variants of the calcium-sensing receptor (CaSR) are associated with increased ER to cytosol calcium gradient. PLoS One. 2013 Nov 14;8(11):e79113.
6) I found some mistakes in legends of figures, for instance in figure 2.
Response: We checked the figure legends. In figure 2 legend, it should be CDN1163.
7) Effects of TG, SERCA inhibitor, on ER Ca content and apoptosis are already well documented. This are not very original results and this does not particularly strengthen the conclusions on the effect of Cd ions on the ER.
Response: We have deleted the conclusion of effects of TG on ER content and apoptosis in the discussion part and revised in the results sections. (Results 2.3, Discussion paragraph 3)
8) I am not convinced by the "representing" western-blot shown in figures (supplementary is better): this blot does not show convincing effects on BiP, PDI and P62. The only effect clearly visible is the effect of Cd on cleaved caspase 3 in coherence with graphs presenting mean values. There is also an expected high effect of Tg on BiP, P62 and cleaved caspases 3.
Response: We revised the description of the results. The results of western blotting showed that Cd slightly increased the expressions of ER stress biomarkers BiP and PDI, as well as apoptosis biomarker cleaved caspase-3, although not as TG so significantly (Fig.3A-E, S3). (Results 2.3)
9) the authors should compare the effects of Cd on SERCA expression with the expression of the ER calcium-release channel IP3R.
Response: Previous studies have proved that Cd induced [Ca2+]c overload with [Ca2+ ]ER decrease, and elevated PLC and IP3 activities and IP3R expression in renal tubular cells (Gu et al., 2018; Zhang et al., 2021; Guo et al., 2022). However, the effects of Cd on Ca2+ reuptake channel of ER have not been clarified. So, in the present study, we focused to explore the role of SERCA, the main Ca2+ reuptake channel of ER in Cd-induced apoptosis of renal tubular cells.
References:
Gu J, Dai S, Liu Y, Liu H, Zhang Y, Ji X, Yu F, Zhou Y, Chen L, Tse WKF, Wong CKC, Chen B, Shi H. Activation of Ca2+-sensing receptor as a protective pathway to reduce Cadmium-induced cytotoxicity in renal proximal tubular cells. Sci Rep. 2018 Jan 18;8(1):1092.
Zhang C, Lin T, Nie G, Hu R, Pi S, Wei Z, Wang C, Li G, Hu G. In vivo assessment of molybdenum and cadmium co-induce nephrotoxicity via causing calcium homeostasis disorder and autophagy in ducks (Anas platyrhyncha).Ecotoxicol Environ Saf. 2021 Dec 25;230:113099.
Guo H, Huang B, Cui T, Chu X, Pu W, Huang G, Xing C, Zhang C. Cadmium exposure induces autophagy via PLC-IP3 -IP3 R signaling pathway in duck renal tubular epithelial cells. Environ Toxicol. 2022 Nov;37(11):2660-2672.
10) the authors should at least discuss known effects of Cd as a blocker of Calcium channels at the plasma membrane.
Response: We discussed the effects of Cd as a blocker of Calcium channels at the plasma membrane in the discussion section (paragraph 2).
In addition, the Ca2+ channels on plasma membrane such as transient receptor potential vanilloid (TRPV) channels could maintain ER Ca2+ homeostasis and protect ER stress-induced apoptosis (Li et al., 2018; Haustrate et al., 2020). However, Cd competites the TRPV5 and TRPV6 channels to block the Ca2+ uptake (Kovacs et al., 2011; 2013; Yang and Shu, 2015).
Reference:
Kovacs, G.; Montalbetti, N.; Franz, M.C.; Graeter, S.; Simonin, A.; Hediger, M.A. Human TRPV5 and TRPV6: Key players in cadmium and zinc toxicity. Cell Calcium 2013, 54, 276–286. PMID: 23968883 DOI: 10.1016/j.ceca.2013.07.003
Kovacs, G.; Danko, T.; Bergeron, M.J.; Balazs, B.; Suzuki, Y.; Zsembery, A.; Hediger, M.A. Heavy metal cations permeate the TRPV6 epithelial cation channel. Cell Calcium 2011, 49, 43–55. PMID: 21146870 DOI: 10.1016/j.ceca.2010.11.007
Yang H, Shu Y. Cadmium transporters in the kidney and cadmium-induced nephrotoxicity Int J Mol Sci. 2015 Jan 9;16(1):1484-94. PMID: 25584611 PMCID: PMC4307315 DOI: 10.3390/ijms16011484
Li Z, Meng Z, Lu J, Chen FM, Wong WT, Tse G, Zheng C, Keung W, Tse K, Li RA, Jiang L, Yao X. TRPV6 protects ER stress-induced apoptosis via ATF6α-TRPV6-JNK pathway in human embryonic stem cell-derived cardiomyocytes. J Mol Cell Cardiol. 2018 Jul;120:1-11. doi: 10.1016/j.yjmcc.2018.05.008. Epub 2018 May 16. PMID: 29758225
Haustrate A, Prevarskaya N, Lehen'kyi V. Role of the TRPV Channels in the Endoplasmic Reticulum Calcium Homeostasis. Cells. 2020 Jan 28;9(2):317. doi: 10.3390/cells9020317. PMID: 32013022
11) editing and English changes are required
Response: We have improved the English writing of the full text.
Round 2
Reviewer 1 Report
The authors have changed the statistical method form Student's t-test to ANOVA and Duncan's multiple comparison. Such changes may have influenced the results of the statistical analyses. However, the figures are completely same as before. I doubt that the authors have just removed the description of Student's t-test. I will believe if the authors show the raw data of the statistical analyses.
THe authors described "Different letters (a, b) indicate statistical significance between control and Cd treatment or between treatments. * " I do not understand the meaning of a or b. Do those mean statistically significant? If compared with control, such a or b should not be placed in the control. The statistical significance should be shown only by *.
Author Response
- The authors have changed the statistical method form Student's t-test to ANOVA and Duncan's multiple comparison. Such changes may have influenced the results of the statistical analyses. However, the figures are completely same as before. I doubt that the authors have just removed the description of Student's t-test. I will believe if the authors show the raw data of the statistical analyses.
Response: Thanks for your comments. We find the raw data as in the PDF file.
- The authors described "Different letters (a, b) indicate statistical significance between control and Cd treatment or between treatments. * " I do not understand the meaning of a or b. Do those mean statistically significant? If compared with control, such a or b should not be placed in the control. The statistical significance should be shown only by *.
Response: Thanks for your comments. We have unified to indicate the statistical significance between control and Cd treatment or between treatments by *. (Figure 2, 3, 5)

Reviewer 2 Report
The authors did not fully answered to my major comments and requests.
1) measurements of resting cytosolic concentration of calcium ions have been done with single emission and single absorption fluorescent probe Fluo3. This must be done with at least Fura-2 ratio metric probe and with calibration in order to avoid arte facts and to compare different cell population.
Response: Fura-2 is usually used as a probe to detect the transient cytosolic Ca2+ level changes as showed in figure 1C-F. Fluo-3 could be applied to measure the cytosolic Ca2+ level at relative steady state (figure 2C-F, G).
I am fully aware that Fura-2 ratiometric (one emission/2excitation wavelength) can be use for measurement of intracellular calcium transients (as it is done in Figure 1 for instance). However I still disagree with the method using measurement of relative steady state cytosolic Ca2+ level with single emmission/single excitation with fluo 3.
Accurate measurements of resting cytosolic Ca2+ level/concentration must be done with ratiometric fluorescent calcium probes such as for instance Fura-2 or Indo 1 (single excitation/2 emission wavelengths) to avoid numerous artefacts due to variation in intracellular concentration of the probe, variation of the cell thickness, etc. Steady-state concentration can then be express and compared between different cell population after calibration in cellulo of the probe fluorescent signal and use of the Grynkiewickz equation (Grynkiewickz et al., 1985).
Fluo3 must be kept for measuring Calcium variations in cells during time and expressed vs F0
Effect of Cd ions on intracellular calcium concentration. The authors should present some recording in conditions of zero calcium in extracellular bath in order to show the independency of responses to calcium entries.
Response: As we known, the intracellular and extracellular Ca2+ levels of cells are in a dynamic equilibrium. If the cells in a zero calcium in extracellular bath condition, Ca2+ will efflux from the cells (intracellular Ca2+ store and cytosolic) to the extracellular to adapt the stress condition. For a long-term bath in the zero calcium medium, the cells will be detached and round. In these statues, it could not identify the role of treatments on the changes of cytosolic Ca2+ levels.
I agree that keeping cells in a zero calcium medium during 24h might be a problem for cell adhesion and survival and prevent to observe long term effects. However I was requesting the control experiment using short term application of Cd (Fig. 1C) in zero calcium medium in order to show if part of the calcium increase is dependent on extracellular calcium. Moreover, this calcium increase is not transient (like with TG) and it would be also important to see the same experiment of Fig. 1C during longer time in order to explore if calcium levels are returning to resting level after short term application of Cd. If not, it could simply mean that cells are dying after Cd application.
3) The authors should show the effect of Tg pretreatment on Cd-dependent calcium transients
Response: We added the result of transient cytosolic Ca2+ level changes of TG treatment (Figure 1 F).
I thank the authors for providing the short term effect of TG on cytosolic Ca2+ level. Interestingly it shows that the Ca2+ level response to Cd is very similar.
However I was requesting in Fig. 1 a short pretreatment with TG in order to deplete ER Ca2+ stores immediately followed by application of Cd. It is a control experiment requested to show that most of calcium mobilized by Cd is released by ER. Same kind of experiment could be down with short pretreatment with 2-APB followed by application of Cd, in order to block IP3R.
4) the authors should show the accumulation of the low affinity Ca probe Mg-Fluo4 in ER, by co-staining with a ER fluorescent probe.
Response: We detected the co-localization of ER Ca2+ content indicator Mag-Fluo-4/AM staining (Green) with ER tracker (Red). (Figure 2A).
I thank the authors for providing the images for co-staining of Mag-Fluo-4/AM staining (Green) with ER tracker (Red). (Figure 2A).
However, these images do not show a clear ER co-staining. Which what kind of fluorescent microscope did the authors performed the images acquisition? The authors need to show convincing images with better spatial resolution. With the images provided by the authors, the distribution seems similar but could be in ER and cytosol as well.
The other points were discussed as requested and corrections have been made in the revised version
Round 3
Reviewer 1 Report
Thank you for presenting all of the data. I calculated statistical significances among groups in Figure 2F. First of all, AOVA is not significant. Fisher's LSD multiple comparison gave the statistical significances between control and TG and between control and TG+cdn1663. If I used Sheffe's multiple comparison, there are no statistical significances between any two groups. Your multiple comparison could be wrong. I suggest re-calculation of statistical significances in all figures. Statistical significances are not so important, but wrong calculations must br corrected.
Figure 2F
ANOVA p=0.0992
Fisher’s LSD (Figure 2F)
|
data1 |
|
|
data2 |
mean1 |
mean2 |
difference |
SD |
P val |
|
control |
|
|
cd |
5363.0000 |
5802.0000 |
439.0000 |
237.8945 |
0.0898 |
|
control |
|
|
cdn1163 |
5363.0000 |
5559.6667 |
196.6667 |
237.8945 |
0.4245 |
|
control |
|
|
cd+cdn1163 |
5363.0000 |
5616.0000 |
253.0000 |
237.8945 |
0.3085 |
|
control |
|
|
TG |
5363.0000 |
5943.6667 |
580.6667 |
237.8945 |
0.0311 |
|
control |
|
|
TG+cdn1163 |
5363.0000 |
6065.3333 |
702.3333 |
237.8945 |
0.0121 |
|
cd |
|
|
cdn1163 |
5802.0000 |
5559.6667 |
242.3333 |
237.8945 |
0.3285 |
|
cd |
|
|
cd+cdn1163 |
5802.0000 |
5616.0000 |
186.0000 |
237.8945 |
0.4494 |
|
cd |
|
|
TG |
5802.0000 |
5943.6667 |
141.6667 |
237.8945 |
0.5626 |
|
cd |
|
|
TG+cdn1163 |
5802.0000 |
6065.3333 |
263.3333 |
237.8945 |
0.2900 |
|
cdn1163 |
|
|
cd+cdn1163 |
5559.6667 |
5616.0000 |
56.3333 |
237.8945 |
0.8168 |
|
cdn1163 |
|
|
TG |
5559.6667 |
5943.6667 |
384.0000 |
237.8945 |
0.1325 |
|
cdn1163 |
|
|
TG+cdn1163 |
5559.6667 |
6065.3333 |
505.6667 |
237.8945 |
0.0550 |
|
cd+cdn1163 |
|
|
TG |
5616.0000 |
5943.6667 |
327.6667 |
237.8945 |
0.1935 |
|
cd+cdn1163 |
|
|
TG+cdn1163 |
5616.0000 |
6065.3333 |
449.3333 |
237.8945 |
0.0833 |
|
TG |
|
|
TG+cdn1163 |
5943.6667 |
6065.3333 |
121.6667 |
237.8945 |
0.6183 |
Scheffe (Figure 2F)
|
data1 |
data2 |
mean1 |
mean2 |
difference |
SD |
P val |
|
control |
cd |
5363.0000 |
5802.0000 |
439.0000 |
237.8945 |
0.6464 |
|
control |
cdn1163 |
5363.0000 |
5559.6667 |
196.6667 |
237.8945 |
0.9805 |
|
control |
cd+cdn1163 |
5363.0000 |
5616.0000 |
253.0000 |
237.8945 |
0.9440 |
|
control |
TG |
5363.0000 |
5943.6667 |
580.6667 |
237.8945 |
0.3695 |
|
control |
TG+cdn1163 |
5363.0000 |
6065.3333 |
702.3333 |
237.8945 |
0.1994 |
|
cd |
cdn1163 |
5802.0000 |
5559.6667 |
242.3333 |
237.8945 |
0.9529 |
|
cd |
cd+cdn1163 |
5802.0000 |
5616.0000 |
186.0000 |
237.8945 |
0.9847 |
|
cd |
TG |
5802.0000 |
5943.6667 |
141.6667 |
237.8945 |
0.9956 |
|
cd |
TG+cdn1163 |
5802.0000 |
6065.3333 |
263.3333 |
237.8945 |
0.9344 |
|
cdn1163 |
cd+cdn1163 |
5559.6667 |
5616.0000 |
56.3333 |
237.8945 |
0.9999 |
|
cdn1163 |
TG |
5559.6667 |
5943.6667 |
384.0000 |
237.8945 |
0.7560 |
|
cdn1163 |
TG+cdn1163 |
5559.6667 |
6065.3333 |
505.6667 |
237.8945 |
0.5098 |
|
cd+cdn1163 |
TG |
5616.0000 |
5943.6667 |
327.6667 |
237.8945 |
0.8534 |
|
cd+cdn1163 |
TG+cdn1163 |
5616.0000 |
6065.3333 |
449.3333 |
237.8945 |
0.6251 |
|
TG |
TG+cdn1163 |
5943.6667 |
6065.3333 |
121.6667 |
237.8945 |
0.9978 |